# The Physical Systems Behind Optimization Algorithms

**Lin F. Yang** *
Princeton University
lin.yang@princeton.edu

**Raman Arora,**
Johns Hopkins University
arora@cs.jhu.edu

**Vladimir Braverman**
Johns Hopkins University
vova@cs.jhu.edu

**Tuo Zhao**†
Georgia Institute of Technology
tourzhao@gatech.edu

## Abstract

We use differential equations based approaches to provide some ***physics*** insights into analyzing the dynamics of popular optimization algorithms in machine learning. In particular, we study gradient descent, proximal gradient descent, coordinate gradient descent, proximal coordinate gradient, and Newton's methods as well as their Nesterov's accelerated variants in a unified framework motivated by a natural connection of optimization algorithms to physical systems. Our analysis is applicable to more general algorithms and optimization problems ***beyond*** convexity and strong convexity, e.g. Polyak-Łojasiewicz and error bound conditions (possibly nonconvex).

## 1 Introduction

Many machine learning problems can be cast into an optimization problem of the following form:

$$x^* = \operatorname*{argmin}_{x \in \mathcal{X}} f(x), \tag{1.1}$$

where $\mathcal{X} \subseteq \mathbb{R}^d$ and $f : \mathcal{X} \to \mathbb{R}$ is a continuously differentiable function. For simplicity, we assume that $f$ is convex or approximately convex (more on this later). Perhaps, the earliest algorithm for solving (1.1) is the vanilla gradient descent (VGD) algorithm, which dates back to Euler and Lagrange. VGD is simple, intuitive, and easy to implement in practice. For large-scale problems, it is usually more scalable than more sophisticated algorithms (e.g. Newton).

Existing state-of-the-art analysis shows that VGD achieves an $\mathcal{O}(1/k)$ convergence rate for smooth convex functions and a linear convergence rate for strongly convex functions, where $k$ is the number of iterations [11]. Recently, a class of Nesterov's accelerated gradient (NAG) algorithms have gained popularity in statistical signal processing and machine learning communities. These algorithms combine the vanilla gradient descent algorithm with an additional momentum term at each iteration. Such a modification, though simple, has a profound impact: the NAG algorithms attain faster convergence than VGD. Specifically, NAG achieves $\mathcal{O}(1/k^2)$ convergence for smooth convex functions, and linear convergence with a better constant term for strongly convex functions [11].

Another closely related class of algorithms is randomized coordinate gradient descent (RCGD) algorithms. These algorithms conduct a gradient descent-type step in each iteration, but only with

respect to a single coordinate. RCGD has similar convergence rates to VGD, but has a smaller overall computational complexity, since its computational cost per iteration of RCGD is much smaller than VGD [10, 7]. More recently, [5, 2] applied Nesterov's acceleration to RCGD, and proposed accelerated randomized coordinate gradient (ARCG) algorithms. Accordingly, they established similar accelerated convergence rates for ARCG.

Another line of research focuses on relaxing the convexity and strong convexity conditions for alternative regularity conditions, including restricted secant inequality, error bound, Polyak-Łojasiewicz, and quadratic growth conditions. These conditions have been shown to hold for many optimization problems in machine learning, and faster convergence rates have been established (e.g. [8, 6, 9, 20, 3, 4]).

Although various theoretical results have been established, the algorithmic proof of convergence and regularity conditions in these analyses rely heavily on algebraic tricks that are sometimes arguably mysterious to understand. To this end, a popular recent trend in the analysis of optimization algorithms has been to study gradient descent as a discretization of gradient flow; these approaches often provide a clear interpretation for the continuous approximation of the algorithmic systems [16, 17]. In [16], authors propose a framework for studying discrete algorithmic systems under the limit of infinitesimal time step. They show that Nesterov's accelerated gradient (NAG) algorithm can be described by an ordinary differential equation (ODE) under the limit that time step tends to zero. In [17], authors study a more general family of ODE's that essentially correspond to accelerated gradient algorithms. All these analyses, however, lack a natural interpretation in terms of physical systems behind the optimization algorithms. Therefore, they do not clearly explain why the momentum leads to acceleration. Meanwhile, these analyses only consider general convex conditions and gradient descent-type algorithms, and are NOT applicable to either the aforementioned relaxed conditions or coordinate-gradient-type algorithms (due to the randomized coordinate selection).

**Our Contribution (I):** We provide novel physics-based insights into the differential equation approaches for optimization. In particular, we connect the optimization algorithms to natural physical systems through differential equations. This allows us to establish a unified theory for understanding optimization algorithms. Specifically, we consider the VGD, NAG, RCGD, and ARCG algorithms. All of these algorithms are associated with **damped oscillator** systems with different **particle mass** and **damping coefficients**. For example, VGD corresponds to a massless particle system while NAG corresponds to a massive particle system. A damped oscillator system has a natural dissipation of its mechanical energy. The decay rate of the mechanical energy in the system is connected to the convergence rate of the algorithm. Our results match the convergence rates of all algorithms considered here to those known in existing literature. We show that for a massless system, the convergence rate only depends on the gradient (force field) and smoothness of the function, whereas a massive particle system has an energy decay rate proportional to the ratio between the mass and damping coefficient. We further show that optimal algorithms such as NAG correspond to an oscillator system near **critical damping**. Such a phenomenon is known in the physical literature that the critically damped system undergoes the fastest energy dissipation. We believe that this view can potentially help us design novel optimization algorithms in a more intuitive manner. As pointed out by the anonymous reviewers, some of the intuitions we provide are also presented in [13]; however, we give a more detailed analysis in this paper.

**Our Contribution (II):** We provide new analysis for more general optimization problems beyond general convexity and strong convexity, as well as more general algorithms. Specifically, we provide several concrete examples: (1) VGD achieves linear convergence under the Polyak-Łojasiewicz (PL) condition (possibly nonconvex), which matches the state-of-art result in [4]; (2) NAG achieves accelerated linear convergence (with a better constant term) under both general convex and quadratic growth conditions, which matches the state-of-art result in [19]; (3) Coordinate-gradient-type algorithms share the same ODE approximation with gradient-type algorithms, and our analysis involves a *more refined* infinitesimal analysis; (4) Newton's algorithm achieves linear convergence under the strongly convex and self-concordance conditions. See Table 1 for a summary. Due to space limitations, we present the extension to the nonsmooth composite optimization problem in Appendix.

Table 1: Our contribution compared with [16, 17].

| [15]/[16]/Ours | VGD | NAG | RCGD | ARCG | Newton |
|---|---|---|---|---|---|
| General Convex | --/--/☑ | ☑/☑/☑ | --/--/☑ | --/--/☑ | --/☑/-- |
| Strongly Convex | --/--/☑ | --/--/☑ | --/--/☑ | --/--/☑ | --/--/☑ |
| Proximal Variants | --/--/☑ | ☑/--/☑ | --/--/☑ | --/--/☑ | --/--/☑ |
| PL Condition | --/--/☑ | --/--/☑ | --/--/☑ | --/--/☑ | --/--/-- |
| Physical Systems | --/--/☑ | --/--/☑ | --/--/☑ | --/--/☑ | --/--/☑ |

Recently, an independent work considered a framework similar to ours for analyzing the first-order optimization algorithms [18]; while the focus there is on bridging the gap between discrete algorithmic analysis and continuous approximation, we focus on understanding the physical systems behind the optimization algorithms. Both perspectives are essential and complementary to each other.

Before we proceed, we first introduce assumptions on the objective $f$.

**Assumption 1.1** ($L$-smooth). There exists a constant $L > 0$ such that for any $x$, $y \in \mathbb{R}^d$, we have $\|\nabla f(x) - \nabla f(y)\| \leq L\|x - y\|$.

**Assumption 1.2** ($\mu$-strongly convex). There exists a constant $\mu$ such that for any $x$, $y \in \mathbb{R}^d$, we have $f(x) \geq f(y) + \langle \nabla f(y), x - y \rangle + \frac{\mu}{2}\|x - y\|^2$.

**Assumption 1.3** . ($L_{\max}$-coordinate-smooth) There exists a constant $L_{\max}$ such that for any $x$, $y \in \mathbb{R}^d$, we have $|\nabla_j f(x) - \nabla_j f(x_{\setminus j}, y_j)| \leq L_{\max}(x_j - y_j)^2$ for all $j = 1, ..., d$.

The $L_{\max}$-coordinate-smooth condition has been shown to be satisfied by many machine learning problems such as Ridge Regression and Logistic Regression. For convenience, we define $\kappa = L/\mu$ and $\kappa_{\max} = L_{\max}/\mu$. Note that we also have $L_{\max} \leq L \leq dL_{\max}$ and $\kappa_{\max} \leq \kappa \leq d\kappa_{\max}$.

## 2 From Optimization Algorithms to ODE

We develop a unified representation for the continuous approximations of the aforementioned optimization algorithms. Our analysis is inspired by [16], where the NAG algorithm for general convex function is approximated by an ordinary differential equation under the limit of infinitesimal time step. We start with VGD and NAG, and later show that RCGD and ARCG can also be approximated by the same ODE. For self-containedness, we present a brief review for popular optimization algorithms in Appendix A (VGD, NAG, RCGD, ARCG, and Newton).

### 2.1 A Unified Framework for Continuous Approximation Analysis

By considering an infinitesimal step size, we rewrite VGD and NAG in the following generic form:
$$x^{(k)} = y^{(k-1)} - \eta\nabla f(y^{(k-1)}) \quad \text{and} \quad y^{(k)} = x^{(k)} + \alpha(x^{(k)} - x^{(k-1)}). \tag{2.1}$$
For VGD, $\alpha = 0$; For NAG, $\alpha = \frac{\sqrt{1/(\mu\eta)}-1}{\sqrt{1/(\mu\eta)}+1}$ when $f$ is strongly convex, and $\alpha = \frac{k-1}{k+2}$ when $f$ is general convex. We then rewrite (2.1) as
$$\left(x^{(k+1)} - x^{(k)}\right) - \alpha\left(x^{(k)} - x^{(k-1)}\right) + \eta\nabla f\left(x^{(k)} + \alpha(x^{(k)} - x^{(k-1)})\right) = 0. \tag{2.2}$$
When considering the continuous-time limit of the above equation, it is not immediately clear how the continuous-time is related to the step size $k$. We thus let $h$ denote the time scaling factor and study the possible choices of $h$ later on. With this, we define a continuous time variable
$$t = kh \quad \text{with} \quad X(t) = x^{(\lceil t/h \rceil)} = x^{(k)}, \tag{2.3}$$
where $k$ is the iteration index, and $X(t)$ from $t = 0$ to $t = \infty$ is a trajectory characterizing the dynamics of the algorithm. Throughout the paper, we may omit $(t)$ if it is clear from the context.

Note that our definition in (2.3) is very different from [16], where $t$ is defined as $t = k\sqrt{\eta}$, i.e., fixing $h = \sqrt{\eta}$. There are several advantages by using our new definition: **(1)** The new definition leads to a unified analysis for both VGD and NAG. Specifically, if we follow the same notion as [16], we need to redefine $t = k\eta$ for VGD, which is different from $t = k\sqrt{\eta}$ for NAG; **(2)** The new definition is more flexible, and leads to a unified analysis for both gradient-type (VGD and NAG) and coordinate-gradient-type algorithms (RCGD and ARCG), regardless of their different step sizes, e.g $\eta = 1/L$ for VGD and NAG, and $\eta = 1/L_{\max}$ for RCGD and ARCG; **(3)** The new definition is equivalent to [16] only when $h = \sqrt{\eta}$. We will show later that, however, $h \asymp \sqrt{\eta}$ is a natural requirement of a massive particle system rather than an artificial choice of $h$.

We then proceed to derive the differential equation for (2.2). By Taylor expansion
$$\left(x^{(k+1)} - x^{(k)}\right) = \dot{X}(t)h + \frac{1}{2}\ddot{X}(t)h^2 + o(h),$$
$$\left(x^{(k)} - x^{(k-1)}\right) = \dot{X}(t)h - \frac{1}{2}\ddot{X}(t)h^2 + o(h),$$
$$\text{and } \eta\nabla f\left[x^{(k)} + \alpha\left(x^{(k)} - x^{(k-1)}\right)\right] = \eta\nabla f(X(t)) + O(\eta h).$$
where $\dot{X}(t) = \frac{dX(t)}{dt}$ and $\ddot{X}(t) = \frac{d^2X}{dt^2}$, we can rewrite (2.2) as
$$\frac{(1+\alpha)h^2}{2\eta}\ddot{X}(t) + \frac{(1-\alpha)h}{\eta}\dot{X}(t) + \nabla f(X(t)) + O(h) = 0. \tag{2.4}$$

Taking the limit of $h \to 0$, we rewrite (2.4) in a more convenient form,

$$m\ddot{X}(t) + c\dot{X}(t) + \nabla f(X(t)) = 0. \tag{2.5}$$

Here (2.5) describes exactly a *damped oscillator system* in $d$ dimensions with

$$
\begin{aligned}
m &:= \tfrac{1+\alpha}{2}\tfrac{h^2}{\eta} && \text{as} && \text{the } \textit{particle mass}, \\
c &:= \tfrac{(1-\alpha)h}{\eta} && \text{as} && \text{the } \textit{damping coefficient}, \\
\text{and } f(x) &&& \text{as} && \text{the } \textit{potential field}.
\end{aligned}
$$

Let us now consider how to choose $h$ for different settings. The basic principle is that **both $m$ and $c$ are finite under the limit** $h, \eta \to 0$. In other words, the physical system is valid. Taking VGD as an example, for which we have $\alpha = 0$. In this case, the only valid setting is $h = \Theta(\eta)$, under which, $m \to 0$ and $c \to c_0$ for some constant $c_0$. We call such a particle system *massless*. For NAG, it can also be verified that only $h = \Theta(\sqrt{\eta})$ results in a *valid* physical system and it is *massive* ($0 < m < \infty, 0 \le c < \infty$). Therefore, we provide a unified framework of choosing the correct time scaling factor $h$.

## 2.2 A Physical System: Damped Harmonic Oscillator

In classic mechanics, the harmonic oscillator is one of the first mechanic systems, which admit an exact solution. This system consists of a massive particle and restoring force. A typical example is a massive particle connecting to a massless spring.

The spring always tends to stay at the equilibrium position. When it is stretched or compressed, there will be a force acting on the object that stretches or compresses it. The force is always pointing toward the equilibrium position. The energy stored in the spring is

$$V(X) := \frac{1}{2}\mathcal{K}X^2,$$

where $X$ denotes the displacement of the spring, and $\mathcal{K}$ is the Hooke's constant of the spring. Here $V(x)$ is called the *potential* energy in existing literature on physics.

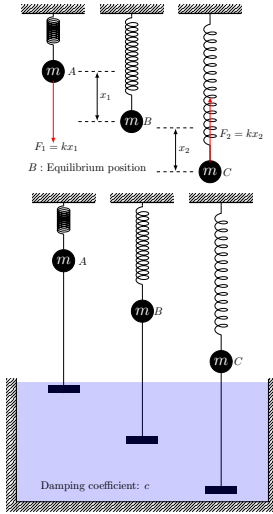

Figure 1: An illustration of the harmonic oscillators: A massive particle connects to a massless spring. (Top) Undamped harmonic oscillator; (Bottom) Damped harmonic oscillator.

When one end of spring is attached to a fixed point, and the other end is attached to a freely moving particle with mass $m$, we obtain a *harmonic oscillator*, as illustrated in Figure 1. If there is no friction on the particle, by Newton's law, we write the differential equation to describe the system:

$$m\ddot{X} + \mathcal{K}X = 0$$

where $\ddot{X} := d^2X/dt^2$ is the acceleration of the particle. If we compress the spring and release it at point $x_0$, the system will start oscillating, i.e., at time $t$, the position of the particle is $X(t) = x_0 \cos(\omega t)$, where $\omega = \sqrt{\mathcal{K}/m}$ is the oscillating frequency.

Such a system has two physical properties: (1) The total energy

$$\mathcal{E}(t) := V(X(t)) + K(X(t)) = V(x_0)$$

is always a constant, where $K(X) := \frac{1}{2}m\dot{X}^2$ is the kinetic energy of the system. This is also called *energy conservation* in physics; (2) The system never stops.

The harmonic oscillator is closely related to optimization algorithms. As we will show later, all our aforementioned optimization algorithms simply simulate a system, where a particle is falling inside a given potential. From a perspective of optimization, the equilibrium is essentially the minimizer of the quadratic potential function $V(x) = \frac{1}{2}\mathcal{K}x^2$. The desired property of the system is to stop the particle at the minimizer. However, a simple harmonic oscillator would not be sufficient and does not correspond to a convergent algorithm, since the system never stops: the particle at the equilibrium has the largest kinetic energy, and the inertia of the massive particle would drive it away from the equilibrium.

One natural way to stop the particle at the equilibrium is adding damping to the system, which dissipates the mechanic energy, just like the real-world mechanics. A simple damping is a force proportional to the negative velocity of the particle (e.g. submerge the system in some viscous fluid)

defined as
$$F_f = -c\dot{X},$$
where $c$ is the *viscous damping coefficient*. Suppose the potential energy of the system is $f(x)$, then the differential equation of the system is,
$$m\ddot{X} + c\dot{X} + \nabla f(X) = 0. \qquad (2.6)$$
For the quadratic potential, i.e., $f(x) = \frac{\mathcal{K}}{2}\|x - x^*\|^2$, the energy exhibits exponential decay, i.e.,
$$\mathcal{E}(t) \propto \exp(-ct/(2m))$$
for *under damped* or nearly *critical damped system* (e.g. $c^2 \lesssim 4m\mathcal{K}$).

For an *over damped system* (i.e. $c^2 > 4m\mathcal{K}$), the energy decay is
$$\mathcal{E}(t) \propto \exp\left( -\frac{1}{2}\left[\frac{c}{m} - \sqrt{\frac{c^2}{m^2} - \frac{4\mathcal{K}}{m}}\right]t\right).$$
For extremely over damping cases, i.e., $c^2 \gg 4m\mathcal{K}$, we have $\frac{c}{m} - \sqrt{\frac{c^2}{m^2} - \frac{4\mathcal{K}}{m}} \to \frac{2\mathcal{K}}{c}$. This decay does not depend on the particle mass. The system exhibits a behavior as if the particle has no mass. In the language of optimization, the corresponding algorithm has linear convergence. Note that the convergence rate does only depend on the ratio $c/m$ and does not depend on $\mathcal{K}$ when the system is under damped or critically damped. The fastest convergence rate is obtained, when the system is critically damped, $c^2 = 4m\mathcal{K}$.

## 2.3 Sufficient Conditions for Convergence

For notational simplicity, we assume that $x^* = 0$ is a global minimum of $f$ with $f(x^*) = 0$. The potential energy of the particle system is simply defined as $V(t) := V(X(t)) := f(X(t))$. If an algorithm converges to optimal, a sufficient condition is that the corresponding potential energy $V$ decreases over time. The decreasing rate determines the convergence rate of the corresponding algorithm.

**Theorem 2.1.** Let $\gamma(t) > 0$ be a nondecreasing function of $t$ and $\Gamma(t) \geq 0$ be a nonnegative function. Suppose that $\gamma(t)$ and $\Gamma(t)$ satisfy
$$\frac{d(\gamma(t)(V(t) + \Gamma(t)))}{dt} \leq 0 \quad \text{and} \quad \lim_{t\to 0^+} \gamma(t)(V(t) + \Gamma(t))) < \infty.$$
Then the convergence rate of the algorithm is characterized by $\frac{1}{\gamma(t)}$.

*Proof.* By $\frac{d(\gamma(t)(V(t)+\Gamma(t)))}{dt} \leq 0$, we have
$$\gamma(t)(V(t) + \Gamma(t)) \leq \gamma(0^+)(f(X(0^+)) + \Gamma(0^+)).$$
This further implies $f(X) \leq V(t) + \Gamma(t) \leq \frac{\gamma(0^+)(f(X(0^+))+\Gamma(0^+))}{\gamma(t)}$. $\qquad \square$

In words, $\gamma(t)[V(t) + \Gamma(t)]$ serves as a Lyapunov function of system. We say that an algorithm is $(1/\gamma)$-*convergent*, if the potential energy decay rate is $\mathcal{O}(1/\gamma)$. For example, $\gamma(t) = e^{at}$ corresponds to linear convergence, and $\gamma = at$ corresponds to sublinear convergence, where $a$ is a constant and independent of $t$. In the following section, we apply Theorem 2.1 to different problems by choosing different $\gamma$'s and $\Gamma$'s.

## 3 Convergence Rate in Continuous Time

We derive the convergence rates of different algorithms for different families of objective functions. Given our proposed framework, we only need to find $\gamma$ and $\Gamma$ to characterize the energy decay.

### 3.1 Convergence Analysis of VGD

We study the convergence of VGD for two classes of functions: **(1)** General convex function — [11] has shown that VGD achieves $\mathcal{O}(L/k)$ convergence for general convex functions; **(2)** A class of functions satisfying the Polyak-Łojasiewicz (PŁ) condition, which is defined as follows [14, 4].

**Assumption 3.1** . We say that $f$ satisfies the $\mu$-PŁ condition, if there exists a constant $\mu$ such that for any $x \in \mathbb{R}^d$, we have $0 < \frac{f(x)}{\|\nabla f(x)\|^2} \leq \frac{1}{2\mu}$.

[4] has shown that the PŁ condition is the weakest condition among the following conditions: strong convexity (SC), essential strong convexity (ESC), weak strong convexity (WSC), restricted secant inequality (RSI) and error bound (EB). Thus, the convergence analysis for the PŁ condition naturally extends to all the above conditions. Please refer to [4] for more detailed definitions and analyses as well as various examples satisfying such a condition in machine learning.

### 3.1.1 Sublinear Convergence for General Convex Function

By choosing $\Gamma(t) = \frac{c\|X\|^2}{2t}$ and $\gamma(t) = t$, we have

$$\frac{d(\gamma(t)(V(t) + \Gamma(t)))}{dt} = f(X(t)) + t\left\langle \nabla f(X(t)), \dot{X}(t) \right\rangle + \left\langle X(t), c\dot{X}(t) \right\rangle$$

$$= f(X(t)) - \langle \nabla f(X(t)), X(t) \rangle - \frac{t}{c} \|\nabla f(X(t))\|^2 \leq 0,$$

where the last inequality follows from the convexity of $f$. Thus, Theorem 2.1 implies

$$f(X(t)) \leq \frac{c\|x_0\|^2}{2t}. \tag{3.1}$$

Plugging $t = kh$ and $c = h/\eta$ into (3.1) and set $\eta = \frac{1}{L}$, we match the convergence rate in [11]:

$$f(x^{(k)}) \leq \frac{c\|x_0\|^2}{2kh} = \frac{L\|x_0\|^2}{2k}. \tag{3.2}$$

### 3.1.2 Linear Convergence Under the Polyak-Łojasiewicz Condition

Equation (2.5) implies $\dot{X} = -\frac{1}{c}\nabla f(X(t))$. By choosing $\Gamma(t) = 0$ and $\gamma(t) = \exp\left(\frac{2\mu t}{c}\right)$, we obtain

$$\frac{d(\gamma(t)(V(t) + \Gamma(t)))}{dt} = \gamma(t)\left(\frac{2\mu}{c}f(X(t)) + \left\langle \nabla f(X(t)), \dot{X}(t) \right\rangle\right)$$

$$= \gamma(t)\left(\frac{2\mu}{c}f(X(t)) - \frac{1}{c}\|\nabla f(X(t))\|^2\right).$$

By the $\mu$-PŁ condition: $0 < \frac{f(X(t))}{\|\nabla f(X(t))\|^2} \leq \frac{1}{2\mu}$ for some constant $\mu$ and any $t$, we have

$$\frac{d(\gamma(t)(V(t) + \Gamma(t)))}{dt} \leq 0.$$

By Theorem 2.1, for some constant $C$ depending on $x_0$, we obtain

$$f(X(t)) \leq C' \exp\left(-\frac{2\mu t}{c}\right), \tag{3.3}$$

which matches the behavior of an extremely over damped harmonic oscillator. Plugging $t = kh$ and $c = h/\eta$ into (3.3) and set $\eta = \frac{1}{L}$, we match the convergence rate in [4]:

$$f(x_k) \leq C \exp\left(-\frac{2\mu}{L}k\right) \tag{3.4}$$

for some constant $C$ depending on $x^{(0)}$.

## 3.2 Convergence Analysis of NAG

We study the convergence of NAG for a class of convex functions satisfying the Polyak-Łojasiewicz (PŁ) condition. The convergence of NAG has been studied for general convex functions in [16], and therefore is omitted. [11] has shown that NAG achieves a linear convergence for strongly convex functions. Our analysis shows that the strong convexity can be relaxed as it does in VGD. However, in contrast to VGD, NAG requires $f$ to be convex.

For a $L$-smooth convex function satisfying $\mu$-PŁ condition, we have the particle mass and damping coefficient as $m = \frac{h^2}{\eta}$ and $c = \frac{2\sqrt{\mu}h}{\sqrt{\eta}} = 2\sqrt{m\mu}$. By [4], under convexity, PŁ is equivalent to quadratic growth (QG). Formally, we assume that $f$ satisfies the following condition.

**Assumption 3.2** . We say that $f$ satisfies the $\mu$-QG condition, if there exists a constant $\mu$ such that for any $x \in \mathbb{R}^d$, we have $f(x) - f(x^*) \geq \frac{\mu}{2} \|x - x^*\|^2$.

We then proceed with the proof for NAG. We first define two parameters, $\lambda$ and $\sigma$. Let

$$\gamma(t) = \exp(\lambda c t) \quad \text{and} \quad \Gamma(t) = \frac{m}{2}\|\dot{X} + \sigma c X\|^2.$$

Given properly chosen $\lambda$ and $\sigma$, we show that the required condition in Theorem 2.1 is satisfied. Recall that our proposed physical system has kinetic energy $\frac{m}{2}\|\dot{X}(t)\|^2$. In contrast to an un-damped system, NAG takes an effective velocity $\dot{X} + \sigma c X$ in the viscous fluid. By simple manipulation,

$$\frac{d(V(t) + \Gamma(t))}{dt} = \langle \nabla f(X), \dot{X}\rangle + m\langle \dot{X} + \sigma c X, \ddot{X} + \sigma c \dot{X}\rangle.$$

We then observe

$$\exp(-\lambda c t)t\frac{d(\gamma(t)(V(t)+\Gamma(t)))}{dt} = \left[\lambda c f(X) + \frac{\lambda c m}{2}\|\dot{X} + \sigma c X\|^2 + \frac{d(V(t)+\Gamma(t))}{dt}\right]$$

$$\leq \left[\lambda c\left(1 + \frac{m\sigma^2 c^2}{\mu}\right)f(X) + \left\langle \dot{X}, \left(\frac{\lambda c m}{2} + m\sigma c\right)\dot{X} + \nabla f(X) + m\ddot{X}\right\rangle\right.$$

$$\left. + \langle X, (\lambda\sigma m c^2 + m\sigma^2 c^2)\dot{X} + m\sigma c\ddot{X}\rangle\right].$$

Since $c^2 = 4m\mu$, we argue that if positive $\sigma$ and $\lambda$ satisfy

$$m(\lambda + \sigma) = 1 \quad \text{and} \quad \lambda\left(1 + \frac{m\sigma^2 c^2}{\mu}\right) \leq \sigma, \tag{3.5}$$

then we guarantee $\frac{d(\gamma(t)(V(t)+\Gamma(t)))}{dt} \leq 0$. Indeed, we obtain

$$\left\langle \dot{X}, \left(\frac{\lambda c m}{2} + m\sigma c\right)\dot{X} + \nabla f(X) + m\ddot{X}\right\rangle = -\frac{\lambda m c}{2}\|\dot{X}\|^2 \leq 0 \quad \text{and}$$

$$\langle X, (\lambda\sigma m c^2 + m\sigma^2 c^2)\dot{X} + m\sigma c\ddot{X}\rangle = -\sigma c\langle X, \nabla f(X)\rangle.$$

By convexity of $f$, we have $\lambda c\left(1 + \frac{m\sigma^2 c^2}{\mu}\right)f(X) - \sigma c\langle X, \nabla f(X)\rangle \leq \sigma c f(X) - \sigma c\langle X, \nabla f(X)\rangle \leq 0$. To make (3.5) hold, it is sufficient to set $\sigma = \frac{4}{5m}$ and $\lambda = \frac{1}{5m}$. By Theorem 2.1, we obtain

$$f(X(t)) \leq C'' \exp\left(-\frac{ct}{5m}\right) \tag{3.6}$$

for some constant $C''$ depending on $x^{(0)}$. Plugging $t = hk$, $m = \frac{h^2}{\eta}$, $c = 2\sqrt{m\mu}$, and $\eta = \frac{1}{L}$ into (3.6), we have that

$$f(x_k) \leq C'' \exp\left(-\frac{2}{5}\sqrt{\frac{\mu}{L}}k\right). \tag{3.7}$$

Comparing with VGD, NAG improves the constant term on the convergence rate for convex functions satisfying PŁ condition from $L/\mu$ to $\sqrt{L/\mu}$. This matches with the algorithmic proof of [11] for strongly convex functions, and [19] for convex functions satisfying the QG condition.

### 3.3 Convergence Analysis of RCGD and ARCG

Our proposed framework also justifies the convergence analysis of the RCGD and ARCG algorithms. We will show that the trajectory of the RCGD algorithm converges weakly to the VGD algorithm, and thus our analysis for VGD directly applies. Conditioning on $x^{(k)}$, the updating formula for RCGD is

$$x_i^{(k)} = x_i^{(k-1)} - \eta\nabla_i f(x^{(k-1)}) \quad \text{and} \quad x_{\setminus i}^{(k)} = x_{\setminus i}^{(k-1)}, \tag{3.8}$$

where $\eta$ is the step size and $i$ is randomly selected from $\{1, 2, \dots, d\}$ with equal probabilities. Fixing a coordinate $i$, we compute its expectation and variance as

$$\mathbb{E}\big(x_i^{(k)} - x_i^{(k-1)}\big|x_i^{(k)}\big) = -\frac{\eta}{d}\nabla_i f\left(x^{(k-1)}\right) \text{ and }$$

$$\text{Var}\left(x_i^{(k)} - x_i^{(k-1)}\big|x_i^{(k)}\right) = \frac{\eta^2(d-1)}{d^2}\left\|\nabla_i f\left(x^{(k-1)}\right)\right\|^2.$$

We define the infinitesimal time scaling factor $h \leq \eta$ as it does in Section 2.1 and denote $\widetilde{X}^h(t) := x^{(\lfloor t/h\rfloor)}$. We prove that for each $i \in [d]$, $\widetilde{X}_i^h(t)$ converges weakly to a deterministic function $X_i(t)$

as $\eta \to 0$. Specifically, we rewrite (3.8) as,

$$\widetilde{X}^h(t+h) - \widetilde{X}^h(t) = -\eta \nabla_i f(\widetilde{X}^h(t)). \tag{3.9}$$

Taking the limit of $\eta \to 0$ at a fix time $t$, we have

$$|X_i(t+h) - X_i(t)| = \mathcal{O}(\eta) \text{ and } \frac{1}{\eta}\mathbb{E}\big(\widetilde{X}^h(t+h) - \widetilde{X}^h(t)\big|\widetilde{X}^h(t)\big) = -\frac{1}{d}\nabla f(\widetilde{X}^h(t)) + \mathcal{O}(h).$$

Since $\|\nabla f(\widetilde{X}^h(t))\|^2$ is bounded at the time $t$, we have $\frac{1}{\eta}\operatorname{Var}\big(\widetilde{X}^h(t+h) - \widetilde{X}^h(t)\big|\widetilde{X}^h(t)\big) = \mathcal{O}(h)$.
Using an infinitesimal generator argument in [1], we conclude that $\widetilde{X}^h(t)$ converges to $X(t)$ weakly as $h \to 0$, where $X(t)$ satisfies, $\dot{X}(t) + \frac{1}{d}\nabla f(X(t)) = 0$ and $X(0) = x^{(0)}$. Since $\eta \leq \frac{1}{L_{\max}}$, by (3.4), we have

$$f(x_k) \leq C_1 \exp\big(-\frac{2\mu}{dL_{\max}}k\big).$$

for some constant $C_1$ depending on $x^{(0)}$. The analysis for general convex functions follows similarly. One can easily match the convergence rate as it does in (3.2), $f(x^{(k)}) \leq \frac{c\|x_0\|^2}{2kh} = \frac{dL_{\max}\|x_0\|^2}{2k}$.

Repeating the above argument for ARCG, we obtain that the trajectory $\widetilde{X}^h(t)$ converges weakly to $X(t)$, where $X(t)$ satisfies

$$m\ddot{X}(t) + c\dot{X}(t) + \nabla f(X(t)) = 0.$$

For general convex function, we have $m = \frac{h^2}{\eta'}$ and $c = \frac{3m}{t}$, where $\eta' = \frac{\eta}{d}$. By the analysis of [16], we have $f(x_k) \leq \frac{C_2 d}{k^2}$, for some constant $C_2$ depending on $x^{(0)}$ and $L_{\max}$.

For convex functions satisfying $\mu$-QG condition, $m = \frac{h^2}{\eta'}$ and $c = 2\sqrt{\frac{m\mu}{d}}$. By (3.7), we obtain $f(x_k) \leq C_3 \exp\big(-\frac{2}{5d}\sqrt{\frac{\mu}{L_{\max}}}\big)$ for some constant $C_3$ depending on $x^{(0)}$.

## 3.4 Convergence Analysis for Newton

Newton's algorithm is a second-order algorithm. Although it is different from both VGD and NAG, we can fit it into our proposed framework by choosing $\eta = \frac{1}{L}$ and the gradient as $L\big[\nabla^2 f(X)\big]^{-1}\nabla f(X)$. We consider only the case $f$ is $\mu$-strongly convex, $L$-smooth and $\nu$-self-concordant. By (2.5), if $h/\eta$ is not vanishing under the limit of $h \to 0$, we achieve a similar equation,

$$\boldsymbol{C}\dot{X} + \nabla f(X) = 0,$$

where $\boldsymbol{C} = h\nabla^2 f(X)$ is the *viscosity tensor* of the system. In such a system, the function $f$ not only determines the gradient field, but also determines a viscosity tensor field. The particle system is as if submerged in an anisotropic fluid that exhibits different viscosity along different directions. We release the particle at point $x_0$ that is sufficiently close to the minimizer 0, i.e. $\|x_0 - 0\| \leq \zeta$ for some parameter $\zeta$ determined by $\nu$, $\mu$, and $L$. Now we consider the decay of the potential energy $V(X) := f(X)$. By Theorem 2.1 with $\gamma(t) = \exp(\frac{t}{2h})$ and $\Gamma(t) = 0$, we have

$$\frac{d(\gamma(t)f(X))}{dt} = \exp\left(\frac{t}{2h}\right) \cdot \left[\frac{1}{2h}f(X) - \frac{1}{h}\left\langle \nabla f(X), (\nabla^2 f(X))^{-1}\nabla f(X)\right\rangle\right].$$

By simple calculus, we have $\nabla f(X) = -\int_1^0 \nabla^2 f((1-t)X)dt \cdot X$. By the self-concordance condition, we have

$$(1 - \nu t\|X\|_X)^2 \nabla^2 f(X) \preceq \nabla^2 f((1-t)X)dt \preceq \frac{1}{(1-\nu t\|X\|_X)^2}\nabla^2 f(X),$$

where $\|v\|_X = \big(v^T\nabla^2 f(X)v\big) \in [\mu\|v\|_2, L\|v\|_2]$. Let $\beta = \nu\zeta L \leq 1/2$. By integration and the convexity of $f$, we have

$$(1-\beta)\nabla^2 f(X) \preceq \int_0^1 \nabla^2 f((1-t)X)dt \preceq \frac{1}{1-\beta}\nabla^2 f(X)$$

$$\text{and } \frac{1}{2}f(X) - \left\langle \nabla f(X), (\nabla^2 f(X))^{-1}\nabla f(X)\right\rangle \leq \frac{1}{2}f(X) - \frac{1}{2}\left\langle \nabla f(X), X\right\rangle \leq 0.$$

Note that our proposed ODE framework only proves a local linear convergence for Newton method under the strongly convex, smooth and self concordant conditions. The convergence rate contains an absolute constant, which does not depend on $\mu$ and $L$. This partially justifies the superior local

convergence performance of the Newton's algorithm for ill-conditioned problems with very small $\mu$ and very large $L$. Existing literature, however, has proved the local quadratic convergence of the Newton's algorithm, which is better than our ODE-type analysis. This is mainly because the discrete algorithmic analysis takes the advantage of "large" step sizes, but the ODE only characterizes "small" step sizes, and therefore fails to achieve quadratic convergence.

## 4 Numerical Simulations

We present an illustration of our theoretical analysis in Figure 2. We consider a strongly convex quadratic program

$$f(x) = \frac{1}{2}x^\top H x, \quad \text{where} \quad H = \left[\begin{array}{cc} 300 & 1 \\ 1 & 50 \end{array}\right].$$

Obviously, $f(x)$ is strongly convex and $x^* = [0,0]^\top$ is the minimizer. We choose $\eta = 10^{-4}$ for VGD and NAG, and $\eta = 2 \times 10^{-4}$ for RCGD and ARCG. The trajectories of VGD and NAG are obtained by the default method for solving ODE in MATLAB.

## 5 Discussions

We then give a more detailed interpretation of our proposed system from a perspective of physics:

**Consequence of Particle Mass** — As shown in Section 2, a massless particle system (mass $m = 0$) describes the simple gradient descent algorithm. By Newton's law, a 0-mass particle can achieve infinite acceleration and has infinitesimal response time to any force acting on it. Thus, the particle is "locked" on the force field (the gradient field) of the potential ($f$) – the velocity of the particle is always proportional to the restoration force acting on the particle. The convergence rate of the algorithm is only determined by the function $f$ and the damping

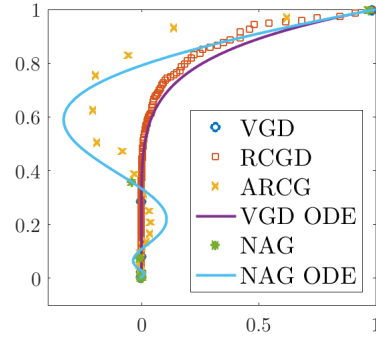

Figure 2: The algorithmic iterates and trajectories of a simple quadratic program.

coefficient. The mechanic energy is stored in the force field (the potential energy) rather than in the kinetic energy. Whereas for a massive particle system, the mechanic energy is also partially stored in the kinetic energy of the particle. Therefore, even when the force field is not strong enough, the particle keeps a high speed.

**Damping and Convergence Rate** — For a quadratic potential $V(x) = \frac{\mu}{2} \|x\|^2$, the system has a exponential energy decay, where the exponent factor depends on mass $m$, damping coefficient $c$, and the property of the function (e.g. PŁ-coefficient). As discussed in Section 2, the decay rate is the fastest when the system is critically damped, i.e, $c^2 = 4m\mu$. For either under or over damped system, the decay rate is slower. For a potential function $f$ satisfying convexity and $\mu$-PŁ condition, NAG corresponds to a nearly critically damped system, whereas VGD corresponds to an extremely over damped system, i.e., $c^2 \gg 4m\mu$. Moreover, we can achieve different acceleration rate by choosing different $m/c$ ratio for NAG, i.e., $\alpha = \frac{1/(\mu\eta)^s - 1}{1/(\mu\eta)^s + 1}$ for some absolute constant $s > 0$. However $s = 1/2$ achieves the largest convergence rate since it is exactly the *critical damping*: $c^2 = 4m\mu$.

**Connecting PŁ Condition to Hooke's law** — The $\mu$-PŁ and convex conditions together naturally mimic the property of a quadratic potential $V$, i.e., a damped harmonic oscillator. Specifically, the $\mu$-PŁ condition

Hooke's constant
Potential Energy of Spring $\rightarrow \frac{\mu}{2}\left\| \frac{\overline{\nabla V}}{\mu} \right\|^2 \geq V(x) \leftarrow$ Potential Energy
Displacement

guarantees that the force field is strong enough, since the left hand side of the above equation is exactly the potential energy of a spring based on Hooke's law. Moreover, the convexity condition $V(x) \le \langle \nabla V(x), X \rangle$ guarantees that the force field has a large component pointing at the equilibrium point (acting as a restoration force). As indicated in [4], PŁ is a much weaker condition than the strong convexity. Some functions that satisfy local PŁ condition do not even satisfy convexity, e.g., matrix factorization. The connection between the PŁ condition and the Hooke's law indicates that strong convexity is not the fundamental characterization of linear convergence. If there is another condition that employs a form of the Hooke's law, it should employ linear convergence as well.

## Footnotes

*Work was done while the author was at Johns Hopkins University. This work is partially supported by the National Science Foundation under grant numbers 1546482, 1447639, 1650041 and 1652257, the ONR Award N00014-18-1-2364, the Israel Science Foundation grant #897/13, a Minerva Foundation grant, and by DARPA award W911NF1820267.

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
