[Supplementary Material]

# A    A Brief Review of Popular Optimization Algorithms

## A.1    Vanilla Gradient Descent Algorithm

A vanilla gradient descent (VGD) algorithm starts from an arbitrary initial solution $x^{(0)}$. At the $k$-th iteration ($k > 0$), VGD takes

$$x^{(k)} = x^{(k-1)} - \eta \nabla f(x^{(k-1)}),$$

where $\eta$ is a properly chosen step size. Since VGD only needs to calculate a gradient of $f$ in each iteration, the computational cost per iteration is usually linearly dependent on $d$. For a $L$-smooth $f$, we can choose a constant step size such that $\eta \leq \frac{1}{L}$ to guarantee convergence.

VGD has been extensively studied in existing literature. [11] show that:

(1) For general convex function, VGD attains a sublinear convergence rate as

$$f(x^{(k)}) - f(x^*) \leq \frac{L\|x^{(0)} - x^*\|^2}{2k} \quad \text{for } k = 1, 2, \dots \tag{A.1}$$

Note that (A.1) is also referred as an iteration complexity of $\mathcal{O}(L/\epsilon)$, i.e., we need $\mathcal{O}(L/\epsilon)$ such that $f(x^{(k)}) - f(x^*) \leq \epsilon$, where $\epsilon$ is a pre-specified accuracy of the objective value.

(2) For a $L$-smooth and $\mu$-strongly convex $f$, VGD attains a linear convergence rate as

$$f(x^{(k)}) - f(x^*) \leq \left(1 - \frac{1}{\kappa}\right)^k \frac{L\|x^{(0)} - x^*\|^2}{2} \quad \text{for } k = 1, 2, \dots \tag{A.2}$$

Note that (A.2) is also referred as an iteration complexity of $\mathcal{O}(\kappa \cdot \log(1/\epsilon))$.

## A.2    Nesterov's Accelerated Gradient Algorithms

The Nesterov's accelerated gradient (NAG) algorithms combines the vanilla gradient descent algorithm with an additional momentum at each iteration. Such a modification, though simple, enables NAG to attain better convergence rate than VGD. Specifically, NAG starts from an arbitrary initial solution $x^{(0)}$ along with an auxiliary solution $y^{(0)} = x^{(0)}$. At the $k$-th iteration, NAG takes

$$x^{(k)} = y^{(k-1)} - \eta \nabla f(y^{(k-1)}) \quad \text{and} \quad y^{(k)} = x^{(k)} + \alpha(x^{(k)} - x^{(k-1)}),$$

where $\alpha = \frac{k-1}{k+2}$ for general convex $f$ and $\alpha = \frac{\sqrt{\kappa}-1}{\sqrt{\kappa}+1}$ for strongly convex $f$. Intuitively speaking, NAG takes an affine combination of the current and previous solutions to compute the update for the two subsequent iterations. This can be viewed as the momentum of a particle during its movement. Similar to VGD, NAG only needs to calculate a gradient of $f$ in each iteration. Similar to VGD, we can choose $\eta \leq \frac{1}{L}$ for a $L$-smooth $f$ to guarantee convergence.

NAG has also been extensively studied in existing literature. [11] show that:

(1) For general convex function, NAG attains a sublinear convergence rate as

$$f(x^{(k)}) - f(x^*) \leq \frac{2L\|x^{(0)} - x^*\|^2}{k^2} \quad \text{for } k = 1, 2, \dots \tag{A.3}$$

Note that (A.3) is also referred as an iteration complexity of $\mathcal{O}(\sqrt{L/\epsilon})$.

(2) For a $L$-smooth and $\mu$-strongly convex $f$, NAG attains a linear convergence rate as

$$f(x^{(k)}) - f(x^*) \leq \left(1 - \sqrt{\frac{1}{4\kappa}}\right)^k \frac{L\|x^{(0)} - x^*\|^2}{2} \quad \text{for } k = 1, 2, \dots \tag{A.4}$$

Note that (A.4) is also referred as an iteration complexity of $\mathcal{O}(\sqrt{\kappa} \cdot \log(1/\epsilon))$.

## A.3    Randomized Coordinate Gradient Descent Algorithm

A randomized coordinate gradient descent (RCGD) algorithm is closely related to VGD. RCGD starts from an arbitrary initial solution $x^{(0)}$. Different from VGD, RCGD takes a gradient descent step only over a coordinate. Specifically, at the $k$-th iteration ($k > 0$), RCGD randomly selects a coordinate $j$ from $1, \dots, d$, and takes

$$x_j^{(k)} = x_j^{(k-1)} - \eta \nabla_j f(x^{(k-1)}) \quad \text{and} \quad x_{\setminus j}^{(k)} = x_{\setminus j}^{(k-1)}.$$

where $\eta$ is a properly chosen step size. Since RCGD only needs to calculate a coordinate gradient of $f$ in each iteration, the computational cost per iteration usually does not scale with $d$. For a $L_{\max}$-coordinate-smooth $f$, we can choose a constant step size such that $\eta \leq \frac{1}{L_{\max}}$ to guarantee convergence.

RCGD has been extensively studied in existing literature. [10, 7] show that:

(1) For general convex function, RCGD attains a sublinear convergence rate in terms of the expected objective value as

$$\mathbb{E}f(x^{(k)}) - f(x^*) \leq \frac{dL_{\max}\|x^{(0)} - x^*\|^2}{2k} \quad \text{for } k = 1, 2, ..... \quad (A.5)$$

Note that (A.5) is also referred as an iteration complexity of $\mathcal{O}(dL_{\max}/\epsilon)$.

(2) For a $L_{\max}$-smooth and $\mu$-strongly convex $f$, RCGD attains a linear convergence rate in terms of the expected objective value as

$$\mathbb{E}f(x^{(k)}) - f(x^*) \leq \left(1 - \frac{\mu}{dL_{\max}}\right)^k \frac{L\|x^{(0)} - x^*\|^2}{2} \quad \text{for } k = 1, 2, ..... \quad (A.6)$$

Note that (A.6) is also referred as an iteration complexity of $\mathcal{O}(dL_{\max}/\mu \cdot \log(1/\epsilon))$.

## A.4  Accelerated Randomized Coordinate Gradient Algorithms

Similar to NAG, the accelerated randomized coordinate gradient (ARCG) algorithms combine the randomized coordinate gradient descent algorithm with an additional momentum at each iteration. Such a modification also enables ARCG to attain better convergence rate than RCGD. Specifically, ARCG starts from an arbitrary initial solution $x^{(0)}$ along with an auxiliary solution $y^{(0)} = x^{(0)}$. At the $k$-th iteration ($k > 0$), ARCG randomly selects a coordinate $j$ from $1, ..., d$, and takes

$$x_j^{(k)} = y_j^{(k-1)} - \eta\nabla_j f(y^{(k-1)}), \quad x_{\setminus j}^{(k)} = y_{\setminus j}^{(k)}, \quad \text{and} \quad y^{(k)} = x^{(k)} + \alpha\left(x^{(k)} - x^{(k-1)}\right).$$

Here $\alpha = \frac{\sqrt{\kappa_{\max}}-1}{\sqrt{\kappa_{\max}}+1}$ when $f$ is strongly convex, and $\alpha = \frac{k-1}{k+2}$ when $f$ is general convex. Similar to RCGD, we can choose $\eta \leq \frac{1}{L_{\max}}$ for a $L_{\max}$-coordinate-smooth $f$ to guarantee convergence.

ARCG has been studied in existing literature. [5, 2] show that:

(1) For general convex function, ARCG attains a sublinear convergence rate in terms of the expected objective value as

$$\mathbb{E}f(x^{(k)}) - f(x^*) \leq \frac{2d\sqrt{L_{\max}}\|x^{(0)} - x^*\|^2}{k^2} \quad \text{for } k = 1, 2, ..... \quad (A.7)$$

Note that (A.7) is also referred as an iteration complexity of $\mathcal{O}(d\sqrt{L_{\max}}/\sqrt{\epsilon})$.

(2) For a $L_{\max}$-smooth and $\mu$-strongly convex $f$, ARCG attains a linear convergence rate in terms of the expected objective value as

$$\mathbb{E}f(x^{(k)}) - f(x^*) \leq \left(1 - \frac{1}{d}\sqrt{\frac{\mu}{L_{\max}}}\right)^k \frac{L\|x^{(0)} - x^*\|^2}{2} \quad \text{for } k = 1, 2, ..... \quad (A.8)$$

Note that (A.8) is also referred as an iteration complexity of $\mathcal{O}(d\sqrt{L_{\max}/\mu} \cdot \log(1/\epsilon))$.

## A.5  Newton's Algorithm

The Newton's (Newton) algorithm requires $f$ to be twice differentiable. It starts with an arbitrary initial $x^{(0)}$. At the $k$-th iteration ($k > 0$), Newton takes

$$x^{(k)} = x^{(k-1)} - \eta(\nabla^2 f(x^{(k-1)}))^{-1}\nabla f(x^{(k-1)}).$$

The inverse of the Hessian matrix adjusts the descent direction by the landscape at $x^{(k-1)}$. Therefore, Newton often leads to a steeper descent than VGD and NAG in each iteration, espcially for highly ill-conditioned problems.

Newton has been extensively studied in existing literature with an additional self-concordant assumption as follows:

**Assumption A.1** . Suppose that $f$ is smooth and convex. We define $g(t) = f(x + tv)$. We say that $f$ is self-concordant, if for any $x \in \mathbb{R}^d$, $v \in \mathbb{R}^d$, and $t \in \mathbb{R}$, there exists a constant $\nu$, which is

independent on $f$ such that we have

$$|g'''(t)| \leq \nu g''(t)^{3/2}.$$

[12] show that for a $L$-smooth, $\mu$-strongly convex and $\nu$-self-concordant, $f$, Newton attains a local quadratic convergence in conjunction. Specifically, given a suitable initial solution $x^{(0)}$ satisfying $\|x^{(0)} - x^*\|_2 \leq \zeta$, where $\zeta < 1$ is a constant depending on on $L$, $\mu$, and $\nu$, there exists a constant $\xi$ depending only on $\nu$ such that we have

$$f(x_{k+1}) - f(x^*) \leq \xi [f(x^{(k)}) - f(x^*)]^2 \quad \text{for } k = 1, 2, \dots \tag{A.9}$$

Note that (A.9) is also referred as an iteration complexity of $\widetilde{\mathcal{O}}(\log\log(1/\epsilon))$, where $\widetilde{\mathcal{O}}$ hides the constant term depending on $L$, $\mu$, and $\nu$. Since Newton needs to calculate the inverse of the Hessian matrix, its per iteration computation cost is at least $\mathcal{O}(d^3)$. Thus, it outperforms VGD and NAG when we need a highly accurate solution, i.e., $\epsilon$ is very small.

## B  Extension to Nonsmooth Composite Optimization

Our framework can also be extended to nonsmooth composite optimization in a similar manner to [16]. Let $g$ be an $L$-smooth function, and $h$ be a general convex function (not necessarily smooth). For $x \in \mathbb{R}^d$, the composite optimization problem solves

$$\min_{x \in \mathbb{R}^d} f(x) := g(x) + h(x).$$

Analogously to [16], we define the force field as the directional subgradient $G(x, p)$ of function $f$, where $G : \mathbb{R}^d \times \mathbb{R}^d \to \mathbb{R}^d$ is defined as $G(x, p) \in \partial f(x)$ and $\langle G(x, p), p \rangle = \sup_{\xi \in \partial f(x)} \langle \xi, p \rangle$, where $\partial f(x)$ denotes the sub-differential of $f$. The existence of $G(x, p)$ is guaranteed by [15]. Accordingly, a new ODE describing the dynamics of the system is

$$m\ddot{X} + c\dot{X} + G(X, \dot{X}) = 0.$$

Under the assumption that the solution to the ODE exists and is unique, we illustrate the analysis by VGD (the mass $m = 0$) under the proximal-PŁ condition. The extensions to other algorithms are straightforward. Specifically, a convex function $f$ satisfies $\mu$-proximal-PŁ if

$$\frac{1}{2\mu} \inf_{p \in S^{d-1}} \|G(x, p)\|^2 \geq f(x) - f(x^*), \tag{B.1}$$

where $x^* = 0$ is the global minimum point of $f$. Slightly different from the definition of the proximal-PŁ condition in [4] involving a step size parameter, (B.1) does not involve any additional parameter. This is actually a more intuitive definition by choosing an appropriate subgradient. Let $\gamma(t) = e^{2\mu t/c}$ and $\Gamma(t) = 0$. For a small enough $\Delta t > 0$, we study

$$\frac{\gamma(t + \Delta t)f(t + \Delta t) - \gamma(t)f(t)}{\Delta t}.$$

By Taylor expansions and the local Lipschitz property of convex function $f$, we have

$$\gamma(t + \Delta t) = \exp\left(\frac{2\mu t}{c}\right)\left(1 + \frac{2\mu}{c}\Delta t\right) + o(\Delta t) \text{ and}$$

$$f(X(t + \Delta t)) = f(X) + \langle \dot{X}, G(X, \dot{X})\langle \Delta t + o(\Delta t) \rangle.$$

Combining the above two expansions, we obtain

$$\gamma(t + \Delta t)f(X(t + \Delta t)) = \exp\left(\frac{2\mu t}{c}\right)\left(f(X) + \frac{2\mu}{c}f(X)\Delta t + \langle \dot{X}, G(X, \dot{X})\rangle \Delta t\right) + o(\Delta t).$$

This further implies

$$\frac{\gamma(t + \Delta t)f(t + \Delta t) - \gamma(t)f(t)}{\Delta t} = \exp\left(\frac{2\mu t}{c}\right)\left(\frac{2\mu}{c}f(X) - \frac{1}{c}\|G(X, \dot{X})\|^2\right) + O(\Delta t).$$

By the $\mu$-proximal-PŁ condition of $f$, we have $\lim_{\Delta t \to 0} \frac{\gamma(t+\Delta t)f(t+\Delta t) - \gamma(t)f(t)}{\Delta t} \leq 0$. The rest of the analysis follows exactly the same as it does in Section 3.1.2.