[Reviews · NeurIPS 2018]

Reviewer 1



The paper presents a continuous-time ODE interpretation of four popular optimization algorithms: Gradient descent, proximal gradient descent, coordinate gradient decent and Newton's method. The four algorithms are all interpreted as damped oscillators with different mass and damping coefficients. It is shown that this ODE formulation can be used to derive the (known) convergence rates in a fairly straight forward manner. Further, the ODE formulation allows to analyze convergence in the non-convex case under the PL-condition. An extension to nonsmooth composite optimization is also discussed. Quality The mathematical analysis seems widely correct. Some aspect on the extension to nonsmooth optimization are unclear to me. The function $G(X,\dot{X})$ must not necessarily be (Lipschitz) continuous. Analyzing the convergence of an ODE with nonsmooth force field requires specific care (e.g., existence & uniqueness of solutions). It seems much more natural to me to analyze nonsmooth functions in a discrete-time setting, rather than in a continuous time setting. The implications of having (potentially) a non-smooth vector field should be carefully analyzed here. Clarity The paper is clearly written, with a straight forward line of argumentation. I would appreciate a brief notation section, introducing all special symbols. Originality The unified interpretation of the different optimization algorithms in one ODE formulation is nice to see. The differentiation to [15] by choosing a different step size is interesting. I would appreciate a further discussion on the impact of different choice. This could for example be investigated in the simulation section. Would the ODE representation of [15] approximate the discrete steps better or worse? I very much like the connection of the PL condition to Hooke's law, although the discussion of this is very brief in the paper. Significance The paper gives some nice insights. However, it seems that the practical relevance remains limited. It is mentioned that the method can potentially help to develop new algorithms, but this step has not been taken. After the authors' feedback: My comments were properly addressed. I have no concerns and recommend to accept the paper.

Reviewer 2



1. Summary This paper described the connection between various optimization algorithms for convex functions and similar variants and differential equations that describe physical systems. This work builds on previous work using differential equations to analyze optimization algorithms. 2. High level paper I think the paper is a bit hard to read from Section 3 onward. Section 3 is packed with proofs of convergence for various types of functions, and new notation is frequently introduced. The contribution of the paper seems novel. And I think this work would be interesting to the optimization community. 3. High level technical Overall I think the biggest things that could be improved are organization of the paper, and the clarity of the writing. Organization: I would spend more time explaining the proof steps in Section 3 and the general connection between optimization algorithms and physical dynamics in Section 5. I think the most interesting contribution of this paper is this optimization-physics connection. This paper will have a lot more impact if you take the time to deeply explain this connection, rather than go through the proofs of each convergence rate. For instance, it would be great to answer the questions: Why must the potential energy decrease for an algorithm to converge? Why is connection between the PL condition and Hooke's law important, or what insight does it give us? Why is the PL condition useful (I know that you cite [4] for examples, but I think you should include some to help with motivation). To have more space for all this I would move Section 4 to the appendix. And possibly move some proofs in Section 3 as well. Clarity: Right now Section 5 assumes a lot of background knowledge in physics. With the additional space I would describe these concepts in more detail, similar to Section 2. A few more specific comments that could improve clarity: - Lines 37-38: Could you be more specific when you say 'analyses rely heavily on algebraic tricks that are sometimes arguably mysterious to be understood'? This would better motivate the method. - Equations above 2.4 (Taylor expansion): Could you derive this quickly? - Line 115: Is it reasonable to require c -> c_0? Doesn't this mean that \eta = O(h) always? Is this reasonable? It would be good to discuss this. - Proof of Theorem 2.1: What does 0^+ mean? - Equation (3.1): How does Theorem 2.1 imply the bound? Do we assume that \gamma(0^+)=1 and f(X(0^+))=0, and do we ignore \gamma(t)? If so, then I see how Theorem 2.1 implies eq. (3.1), but otherwise I don't... - Line 177: It only implies this if m=0 or X^{..}=0, is this true? 4. Low level technical - Line 1: 'differential equations based' -> 'differential equations-based' - Line 17: 'that achieves' -> 'that it achieves' - Line 39: 'recently attract' -> 'recently have attracted' - Line 45: 'are lack of link' -> 'lack a link' - Line 90: 'We starts with' -> 'We start with' - Line 124: I would remove the word 'mechanic' because this quantity is really the potential energy, as you describe below, and I think 'mechanic' creates confusion. - Line 156: 'independent on' -> 'independent of' - Line 167: 'strongly' -> 'strong' - Line 202: 'suffice' -> 'sufficient' - Line 270: 'As have been' -> 'As has been' 5. 1/2 sentence summary Overall, while the paper is somewhat unclear, I think the novelty of the contribution, and its impact, justifies this paper being accepted. Post-Rebuttal ---------------------- The authors do a very thorough job of addressing my review, but are slightly sloppy answering one of my questions: the response on lines 30-32 is a bit sloppy because: (a) they define t=kh in the paper, but in the response they say k=th, (b) in the response they say (x^{(k-1)} - x^{(k)}) = X(t + h) - X(t), but it must actually be equal to X(t - h) - X(t), furthermore, the initial equation (x^{(k-1)} - x^{(k)}) appears in none of the lines directly above equation (2.4) in the paper. For me, I think this paper is worth accepting, but it is so dense that I think the authors need to move some of the proofs out of the main paper and into the supplementary to properly explain all of the concepts. Ultimately, because of their detailed response in the rebuttal, I still believe the paper should be accepted.

Reviewer 3



Summary: This paper contributes to a growing body of work that links optimization algorithms and associated gradient flows to certain continuous time ODEs. In particular, this paper reformulates vanilla and accelerated gradient descent, Newton's method and other popular methods in terms of damped oscillator systems with different masses and damping coefficients. The potential energy of the system serves as a Lyapunov function whose decrease rate can then be related to convergence rates of different iteration schemes. Evaluation: Damped oscillator systems are very well studied; the notion of potential energy is clearly related to Lyapunov methods which have been previously used for analyzing optimization algorithms. Despite the fact that closely related analyses have appeared recently, the paper is overall well presented and the unifying link to the dynamics of a natural system is insightful. - The paper is a bit dense and hard to read at places. - The connection to "A Lyapunov Analysis of Momentum Methods in Optimization" is not fully explained. - The role of ODE integrator that connects the continuous time dynamics to the actual algorithm is not made clear.